# Biocompatibility of Lithium Disilicate and Zirconium Oxide Ceramics with Different Surface Topographies for Dental Implant Abutments

**DOI:** 10.3390/ijms22147700

**Published:** 2021-07-19

**Authors:** Susanne Jung, Marco Maria Moser, Johannes Kleinheinz, Arndt Happe

**Affiliations:** 1Research Unit Vascular Biology of Oral Structures (VABOS), Department of Cranio-Maxillofacial Surgery, University Hospital Muenster, 48149 Münster, Germany; Susanne.Jung@ukmuenster.de (S.J.); marcomariamoser@gmail.com (M.M.M.); Johannes.Kleinheinz@ukmuenster.de (J.K.); 2Private Practice, 48165 Muenster, Germany; 3Center of Dentistry, Department of Prosthetic Dentistry, University of Ulm, 89081 Ulm, Germany

**Keywords:** dental implants, gingivofibroblasts, surface roughness, lithium disilicate, zirconium dioxide, titanium

## Abstract

Gingivafibroblasts were cultured on lithium disilicate, on zirconia dioxide, and on titanium with two different surface roughnesses (0.2 µm and 0.07 µm); Proliferation (MTT), Living/Dead staining, cytotoxicity (LDH), proliferation (FGF2), and inflammation (TNFα) were analyzed after 1 day and 21 days. Furthermore, alteration in cell morphology (SEM) was analyzed. The statistical analysis was performed by a Kruskal–Wallis test. The level of significance was set at *p* < 0.05. There were no distinct differences in cellular behavior between the tested roughness. There were slight differences between tested materials. Cells grown on zirconia dioxide showed higher cytotoxic effects. Cells grown on lithium disilicate showed less expression of TNFα compared to those grown on zirconia dioxide or titanium. These effects persisted only during the first time span. The results indicate that the two tested high-strength ceramics and surface properties are biologically suitable for transmucosal implant components. The findings may help clinicians to choose the most appropriate biomaterial as well as the most appropriate surface treatment to use in accordance with specific clinical dental applications.

## 1. Introduction

All bone-level implants need a transmucosal component to be functional. This component pierces through the oral mucosa and connects the osseointegrated implant with the suprastructure. The transmucosal component is traditionally called an abutment and serves as the three-dimensional transition from the geometric implant diameter to the anatomical emergence profile of the crown. 

Depending on the restorative concept of the implant system and the insertion depth of the implant, the restorative material will be in intimate contact with the peri-implant tissues and the oral cavity [1]. Hence, the interaction of an implant-supported restoration with the oral environment requires the maintenance of an effective biological seal of soft peri-implant tissue and surfaces with low plaque adhesion [2,3,4]. A lack of or inadequate adhesion of the peri-implant soft tissue carries the risk of bacterial penetration and subsequent peri-implantitis and progressive loss of the hard and soft tissue [5]. The ideal abutment is smooth enough to prevent excessive biofilm formation and rough enough to allow for adhesion of the peri-implant soft tissues. That is why an important parameter for the biological effect of materials used for implant restorations and implant abutments is the surface roughness of the component.

Because rough surfaces are difficult to clean and result in a rapid regrowth of the biofilm, the transmucosal abutment is usually polished to nanoroughness [6,7]. The roughness (Ra) of machined titanium stock abutments of different implant manufacturers varies between 0.15 μm and 0.24 μm [8]. Commercially available Brånemark standard titanium abutments (Nobelpharma), for example, were shown to have a surface roughness of approximately Ra = 0.2 µm [9,10]. 

Following abutment installation surgery, fibroblasts play a leading role during the healing process of the oral mucosa towards the material of the abutment, especially in inhibiting epithelial down growth [11]. After a fibrin clot adheres to the surface of the abutment [12], the epithelium starts proliferating over this bridge to the surface of the abutment and starts growing down in the coronal-apical direction until it reaches the underlying connective tissue [13]. Together with the junctional epithelium (around 2 mm), the connective tissue (around 1–1.5 mm) forms the so-called biological width that forms the biological seal that protects the underlying bone towards the oral cavity to prevent bacterial penetration [14].

In the early healing of the connective tissue wound, the formation and adhesion of the fibrin clot to the implant or abutment surface leads to connective tissue cells on the implant’s surface, transforming the clot into granulation tissue [15]. After tissue maturation, the connective tissue portion, located between the epithelium and the marginal bone, is poor in cells and in vascular structures but rich in collagen fibers. These fibers run more or less parallel to the surface of the abutment. This is a major difference between the anatomy of natural teeth, where the dento-gingival collagen fibers firmly insert into the cementum and the bone, serving as a barrier to epithelial migration and invasion [16]. Therefore, the connective tissue adhesion at the transmucosal component (abutment) has a poor mechanical resistance compared to that of natural teeth [17]. This lack of mechanical resistance can potentially endanger the prognosis of oral implants. Tearing the connective tissue/abutment interface could induce the apical migration of the epithelium, accompanied by soft tissue recession of pocket formation and by bone resorption [3]. This underlines the importance of the connective tissue cell-adhesion towards abutment materials.

The anatomy of this biological seal may be affected by the nature of the abutment-regarding material [18], macro-geometry [19], and surface [20]. There is scientific evidence that smooth surfaces favor apical migration of the epithelium and do not favor cell adhesion [1,15,21,22,23] and that the length of the connective tissue zone is higher on rough surfaces than for smooth surfaces, with an inverse relationship for the length of the epithelium [24].

For many years, abutments were made of titanium and were delivered by the industry [8,25], but today they can be made of different materials and manufactured in several ways by local dental laboratories [26,27]. Abutments were mostly produced from only one material (one-piece abutment), or they were made from two parts (two-piece abutment). 

In 1994, Wohlwend introduced an all-ceramic abutment made of yttria-stabilized zirconia poly crystal (Y-TZP, zirconium dioxide) [20]. These zirconia abutments had advantages regarding the overall aesthetic appearance of the peri-implant soft-tissue and restoration [27]. In the early stage of the clinical use of these abutments, the indication of these components was limited to the anterior region. These abutments proved to have good technical and biological outcomes but carry specific risks, like, for example, fracture, that are linked to the nature of the material [28,29]. 

Current restorative concepts involve novel computer-aided design (CAD)/computer-assisted manufacture (CAM) technology in order to fabricate individual abutments with the desired emergence profile. The ceramic part is bonded on an industrially prefabricated secondary metallic component (titanium base) in order to enhance the strength of the component [30,31] and to avoid damage in the interface between the ceramic abutment and the titanium implant by micromovements [32,33]. 

The second generation of high-strength lithium disilicate ceramics (also known as “IPS e max” (Ivoclar Vivadent, Germany)) was introduced in 2005 and proved to have good clinical performance for single crowns and short span partial fixed dental prostheses on natural teeth [34].

Due to its high natural optical appearance and high flexural strength, the modern lithium disilicate materials also allow for manufacturing an implant-borne monolithic all-ceramic suprastructure, which is directly bonded on a secondary metal component (titanium base), yielding a screw-retained restoration without a traditional abutment [35]. This concept changes the indication of the material from a material that was used supra-gingivally to a material that is also used sub-gingivally.

Thus, while the material has been used clinically as an abutment material in close contact with the mucosa for some time now, the number of studies that have investigated its biocompatibility with these specific human cell lines is rather small [36,37,38,39]. Furthermore, the influence of surface modifications, which are common for abutments, on the cell response is interesting. That is why it is important to include this factor in the biocompatibility testing. Some studies have reported on surface modification [18,37,38], while some have not [36,39]. Only very few studies have compared the new material LS with the common abutment materials titanium and/or zirconium in cell culture, also considering surface modifications by testing different surface roughness [38].

The aim of this cell culture study was to test the cell response and biocompatibility of two different surface roughnesses on two different high-strength ceramics against a test group of the so-called golden standard machined titanium. 

## 2. Results

### 2.1. Cell Viability

Cell viability among tested specimens at day 1 and at day 21 is shown in Figure 1. Changes between day 1 and day 21 were significant, with a *p*-value of 0.009 for MTT and LDH release and with a *p*-value of at least 0.014 for TNFα expression. There was no significance for FGF2 expression, with the exception of LiS_2_, with a roughness of 0.2 µM, and titanium, with a roughness of 0.07 µm, with a *p*-value of 0.009 and 0.044, respectively.

For all tested specimens, no distinctive features were observed. Proliferation rate increased over time. The factor of increase varied between tested specimens (Figure 1 MTT). No serious cytotoxic effects were observed. Detection of lactate dehydrogenase (LDH) in the culturing medium was lower or the same as in the control, except for zirconia dioxide with a surface roughness of 0.07 µm. Here, slightly cytotoxic effects within the first 24 h were observable, with a factor of 1.35 (Figure 1 LDH). No distinct features in expression of FGF2 were observed, except for zirconia dioxide with a roughness of 0.2 µm. Here, the strongest decrease over time was observable. 

Expression rate remained constant or decreased slightly over time in other tested specimens (Figure 1 FGF2). Furthermore, no hints of possible inflammation reaction were observable. The expression of TNFα in tested specimens was lower or similar to the control, except for titanium. Here, on both roughnesses, a higher expression rate during the first 24 h was observable (Figure 1 TNFα). For interpretation, a strong limitation of the study by choosing TNFα as a single representative for cytokines should be considered. 

### 2.2. Data Analysis with Focus on Roughness and Material

Considering surface roughness and used material, results were rearranged, and the summarized results are shown in Figure 2. Concerning material, changes between day 1 and day 21 were significant, with a *p*-value of 0.017 for MTT and with a *p*-value of at least 0.009 for LDH release. For FGF2 expression, observed changes were only significant in the titanium group, with a *p*-value of 0.036. For TNFα expression, observed changes were significant in the LiS_2_ group and titanium group, with a *p*-value of 0.009, and not in the ZrO_2_ group. Concerning roughness, changes between day 1 and day 21 were significant, with a *p*-value of 0.009 for MTT, for LDH release, and for TNFα expression. Observed differences for FGF2 expression were only significant in the 0.2 µm group, with a *p-*value of 0.009.

The results rearranged by tested material reflect the results outlined above (Figure 2). In a direct comparison, lithium disilicate showed less negative effects on HGF-1 cells than zirconia dioxide that induced a higher release of LDH during the first 24 h or than titanium with a higher expression rate of TNFα during the first 24 h. Surface roughness seemed to exert no effects directly. Cell viability of HGF-1 cells was similar between both roughnesses (Figure 2). Proliferation rate increased with nearly the same factor, 1.14, with a smoother surface (0.07 µm) and by 1.15 with a rougher surface (0.2 µm). Observed cytotoxic effects on smoother surfaces (0.07 µm) were material-dependent (zirconia dioxide). The expression of FGF2 and TNFα were nearly identical between both roughnesses (Figure 2).

### 2.3. Cell Morphology

Alterations in cells’ arrangement and morphology were summarized for control and for lithium disilicate in Figure 3 and for zirconia dioxide and for titanium in Figure 4. Cells adhered to all tested rougnesses and materials. Pictures of day 1 were presented; no further morphological prediction was possible after day 7, because cells overgrew the surfaces. Living/dead staining confirmed results from the MTT assay. None or less dead cells were observed. In SEM analysis, no distinct differences in cell morphology between surface roughness or used material was observed. The cells lay planar with few extensions on the surface, in relation to the control.

## 3. Discussion

This cell culture study showed a comparable cell response and biocompatibility of all three tested materials and both surface roughnesses. There was no difference between the two tested groups of high strength ceramics and no difference between those groups when compared to the golden standard, titanium. Moreover, the different surface roughness did not influence the biologic response.

These findings are of relevance, as lithium disilicate ceramics were originally developed for conventional restorative dentistry and just recently have been used as an abutment material in implant dentistry. Although the material is now used in intimate contact to the peri-implant soft tissues, only scarce evidence on the biocompatibility of this high-strength ceramic with respect to the human mucosa is available.

Clinically, the adhesion of connective tissue cells to the abutment surface plays a leading role in early healing and in the formation of the biologic width around implants [15,17], which is why we investigated HGF-1 cell response. A previous study looked at the proliferation and attachment behavior of HGF-1 on zirconia and a titanium alloy with different surface topographies (machined, polished, sand-blasted) [40]. The authors assessed the cell response using cell count and vinculin expression by immunocytochemistry and found significant differences between the groups. They found higher proliferation rates of HGF-1 on zirconia than on the titanium alloy and higher cell spreading on polished and machined surfaces than on sand-blasted ones. Rough surfaces provided favorable properties in terms of cellular adhesion of fibroblasts but not of epithelial cells. They concluded that the HGF-1 cell response is influenced by both the material and surface topography. These results are not in line with the findings of our study; however, different surface treatments, a different study design, and different methods for assessment were used.

As described in the introduction, very few studies have compared lithium disilicate with the common abutment materials titanium and/or zirconium in cell culture, while also considering the aspect of surface roughness [38].

One of the few that included this new material was a cell culture that used single-cell force spectroscopy to evaluate cell adhesion of living gingival fibroblasts on different materials and surface roughness [18]. The authors did not find any difference between the materials, but they found significant differences between the different surface roughnesses, showing the strongest cell attachment on machine-like surface roughness. Due to the different methodology, these findings ca not be compared with the results of the present study.

Another cell culture experiment compared the biological response of epithelial tissue cultivated on lithium disilicate and zirconia, yielding an assessment of the influence on gingival would closure of these materials [38]. The authors looked at cell migration and found the best migration on zirconia. Cell adhesion was better on rough and polished lithium disilicate than on the zirconia specimen. Increasing surface roughness led to better cell adhesion. However, the roughness of the specimen varied from Ra 0.01 µm to 2.53 µm. Moreover, the experiment used a chicken epithelium model and not human gingivofibroblasts. Hence, these findings cannot be compared.

A recent animal study tested different abutments luted to a titanium base against one-piece titanium abutments, among them being zirconia and lithium disilicate. Histomorphometric results showed no difference in the anatomy of the biological width between zirconia and lithium disilicate in the animal experiment. These results are in line with our results, as the cell response of HGF-1 on both materials did not show any statistically significant difference. The same animal study showed a statistically significant difference only for the length of the junctional epithelium between zirconia and titanium. Hence, the only difference was found for the epithelium but not for the connective tissue.

There is strong evidence in literature that abutment material and its mechanical, physical, and chemical modification influence fibroblast response in vitro, especially in the earlier phase of contact with the abutment surface [41]. Therefore, the cell response of fibroblasts and the nature of connective tissue healing towards abutment materials is frequently assessed in research and remains a highly relevant topic in implant dentistry [42,43]. Even if the differences we found in this cell culture were not statistically significant, some trends are evident in the results. In direct comparison, lithium disilicate showed less adverse effects for HGF-1 cells than zirconia dioxide that induced a higher release of LDH during the first 24 h or than titanium with a higher expression rate of TNFα during the first 24 h. This might be an indication that this high-strength ceramic, which already showed good results regarding clinical performance as a restorative material, might be also suitable as an alternative to zirconia for the fabrication of an all-ceramic transmucosal implant component.

A recent review on the effects of abutment characteristics on peri-implant soft tissue health concluded that surface topography did not have a significant influence on peri-implant inflammation but the material obviously did. The results of their meta-analysis of 13 clinical studies indicated that zirconia abutments experienced less increase in bleeding on probing (BOP) values over time when compared to titanium abutments [43]. As the BOP is one of the major clinical parameters for peri-implant tissue health [44], these results underline the clinical relevance of the tissue response towards abutment materials.

If the high-strength ceramic lithium disilicate prooves to be an alternative to zirconia, not only regarding its biomechanical properties, but also from a biological standpoint, clinicians could benefit from some advantages of this material. In contrast to zirconia, which can only be processed using a CAD/CAM workflow, in addition, lithium disilicate can also be pressed into the desired shape. Moreover, the material has different optical properties [45]. As described in the introduction, these material characteristics lead to a novel approach for the fabrication of implant borne supra-structures, yielding a screw-retained restoration without a traditional abutment [35], which simplifies the treatment.

The two different surface roughnesses that were tested in this experiment did not exhibit any effect on HGF-1. This is in concordance with the findings of the meta-analysis on the clinical effect of surface modification of abutment conducted by Canullo et al. [46]. They reported that surface topography of abutments did not affect the biocompatibility in the short term and reported contrasting results regarding longer follow-up periods. Hence, these effects may not have been detectable in our particular cell culture set up. Moreover, the clinical effect of surface modifications on the soft tissue response may also be influenced by the biofilm formation that may be related to the surface roughness. However, in this particular experiment, two surfaces below the threshold of 0.2 µm were chosen [6,7]. There is a consensus that implant abutments should have a surface roughness of around 0.2 µm and that implant manufacturers deliver their transmucosal components with a roughness between 0.15 µm and 0.24 µm [7,9,10]. That is because polishing the surface beyond a certain threshold roughness does not influence supra- and submosal composition of the biofilm. Hence, it can be speculated that the tested surfaces in this experiment may not be prone to excessive biofilm formation in clinical use.

At the same time, a certain roughness seems to be reasonable for sufficient cell adhesion. However, within the limitations of this in vitro study, this particular effect could not be seen on the two tested surfaces. Generally, it is a typical limitation of in vitro cell studies that they include relatively simple 2D culture models, which lack the complexity required to draw relevant conclusions [47]. Furthermore, as previously stated, various other factors like, for example, biofilm formation, are known to be important criterions for the clinical tissue response of abutment surfaces and materials [6,8,48].

From this in vitro experiment, it may be concluded that regarding biocompatibility and cell response, the two tested high-strength ceramics and surface properties are biologically suitable for transmucosal implant components.

However, randomized clinical trials have to proove the soft tissue response and clinical performance of lithium disilicate material in the transmucosal area of implant restorations over time.

## 4. Materials and Methods

### 4.1. Study Design

The study evaluated the cell viability of human gingiva fibroblasts (HGF-1) on titanium, lithium, and zirconia with two different surface roughnesses (0.07 µm and 0.2 µm). The cell viability was assessed based on cell vitality, proliferation, and cytotoxicity, whereas the expression of FGF2 and TNFα was assessed based on the protein expression analysis. Cell morphology was recorded based on SEM analysis.

### 4.2. Preparation of Specimen Disks

For this study, in total, 96 disks of titanium, lithium disilicate, and zirconia dioxide were produced, measuring two kinds of roughness, and they were formed into 100 mm diameter and 3 mm thickness. The disks were then processed to get two types of surface roughness. Thus, each of the materials was manufactured with 0.07 µm RA and 0.2 µm RA (arithmetical mean deviation of the assessed profile). To ensure repeatable results, all discs were processed mechanically with grinding wheels of different grain sizes. The parameters affecting specimen preparations were preset. There were 5 parameters to monitor (a) the sort of grinding wheel, (b) the direction of rotation (forwards, backwards), (c) the lubricant, (d) the contact pressure, and (e) the time of application. To get efficient removal, the specimen was placed around the center. Titanium was processed with grinding wheels MD-Gekko 320–1000 plus MD-Largo (Struers GmbH; Willich, Germany). Lithium disilicate and zirconia dioxide disks were treated by MD-Piano 120–600 (Struers GmbH; Willich, Germany). The grinding protocol can be found in Table 1.

Following polishing and before each individual experiment, disks were cleaned and disinfected in multiple steps, starting by scrubbing each specimen disk for 15 s with an electric toothbrush in a 1% Liquinox solution (Alconox, White Plains, NY, USA). Subsequently, the samples were rinsed with distilled water and placed in cell culture plates (Greiner Bio One; Bad Nenndorf, Germany) for a three-step ultrasonic treatment. In the first step, 1% Liquinox solution was applied for 5 min, which was followed by 2 min in distilled water and concluded by a 5 min treatment step in 70% isopropyl alcohol. After application of the isopropyl alcohol, disks were washed twice with sterile water and dried under sterile conditions. Finally, disks were placed into sterile 24-well cell culture plates and were rewashed before using. This included a short storage of 20 min in 70% isopropyl alcohol, a two-fold rinse with distilled water, and followed by a drying period.

### 4.3. Human Cell Culture

Human gingival fibroblasts (HGF-1; ATCC^®^ CRL-2014™; LGC Standards GmbH, Wesel, Germany) were delivered cryopreserved. Thawing of cells was performed according to the manufacturer’s protocols. Culture techniques of human gingiva fibroblasts was performed as described previously [49]. The handling of all human samples strictly adhered to the “Declaration of Helsinki”.

Human gingiva fibroblast cells were cultivated in DMEM high glucose (#21969035; Gibco™, ThermoFisher Scientific; Wesel, Germany) and DME/Hams nutrition mixture F12 (Sigma-Aldrich, Hamburg, Germany). Culturing media was supplemented with 10% bovine calf serum, 1% Amphotericin B, and 1% Penicillin (10,000 U/mL)/Streptomycin (10,000 g/mL) (Biochrom Merck, Berlin, Germany). The cells were cultivated in a 5% CO_2_ humidified atmosphere at 37 °C, being fed every 2–3 days and passaged with 10,000 cells/cm² after reaching 90% confluence.

### 4.4. Main Cell Culture

Cells were cultured on various titanium, lithium disilicate, and zirconia dioxide disks; the groups were (1) titanium implants with 0.07 µm and 0.2 µm surface roughness (Ti—0.07 and Ti—0.2); (2) lithium disilicate with 0.07 µm and 0.02 µm surface roughness (Li—0.07 and Li—0.2); (3) zirconia dioxide with 0.07 µm and 0.02 µm surface roughness (Zr—0.07 and Zr—0.2). The samples were placed in 24-well cell culture plates (Greiner Bio One; Bad Nenndorf, Germany). The culturing medium was replaced every 2–3 days during the cell culture study. In addition, cells for the control group were cultivated as a monolayer in 24-well cell culture plates. Samples were analyzed 1 day and 21 days after the beginning of the experiment. The cell culture procedure was repeated three times.

### 4.5. Cell Viability

Proliferation rate was estimated with an in-house MTT assay, which determines the metabolic activity of vital cells. The conversion of the yellow thiazolyl blue tetrazolium bromide (0.5 mg/mL; Sigma-Aldrich, Hamburg, Germany) to the purple formazan was measured at a wavelength of 570 nm. Cytotoxic effects were determined with the CytoTox 96^®^ Non-Radioactive Cytotoxicity Assay (Promega, Walldorf, Germany). All assays were performed according to manufacturer’s protocols and performed in triplicates.

The qualitative analysis of cell viability was performed via fluorescein diacetate/propidium iodide (FDA/PI) staining, where FDA (Sigma Aldrich; Hamburg, Germany) stained viable cells green, and PI (Fluka; Hamburg, Germany) stained necrotic and apoptotic cell nuclei red.

### 4.6. Protein Expression Analysis

For protein expression analysis, cells were lysed with the Pierce™ IP Lysis Buffer (ThermoFisher Scientific, Wesel, Germany) according to the manufacturer’s protocol. The supernatant was frozen at −80 °C for subsequent assays. To determinate secreted proteins, part of the culturing medium was taken before lysis and frozen at −80 °C. Quantification of protein determination was performed with the Pierce™ BCA Protein Assay (ThermoFisher Scientific, Wesel, Germany) according to the manufacturer’s protocol. The protein expression of fibroblast growth factor 2 (FGF-2) and tumor necrosis factor alpha (TNF-a) were estimated with ELISA kits from PromoCell (PK-EL-60240, PK-EL-63707; Heidelberg, Germany). ELISAs were performed according to the manufacturer’s protocol. The µQuant reader (BioTek, Bad Friedrichshall, Germany) was used for protein determination and ELISA.

### 4.7. SEM Analysis

For SEM analysis, cells cultures were fixed in glutaraldehyde (4% in phosphate buffer, pH 7.4), washed with PBS, and dehydrated in a graded ethanol series (30%, 50%, 70%, 90%, 96%, and absolute). Then, the samples were subjected to critical-point drying with liquid CO_2_ according to the standard procedure. Subsequently, the samples were mounted on an aluminum specimen holder by using conductive adhesive tabs (Plano, Wetzlar, Germany) and sputter-coated with a gold layer having a thickness of approximately 15 nm. Imaging was performed with an S800 SEM (Hitachi Ltd., Tokyo, Japan) at the Institute of Medical Physics and Biophysics (Division: Electron Microscopy, University of Muenster).

### 4.8. Statistical Analysis

The statistical software SPSS version 27 (IBM, Ehningen, Germany) was used for statistical analysis. By the chosen experimental design and the chosen testing parameters, the study was strongly limited. Due to the non-normal distribution of the data, independent parameters, and small sample size, statistical analysis by using the Kruskal–Wallis test was used. The level of significance was set at *p* < 0.05.

## Figures and Tables

**Figure 1 ijms-22-07700-f001:**
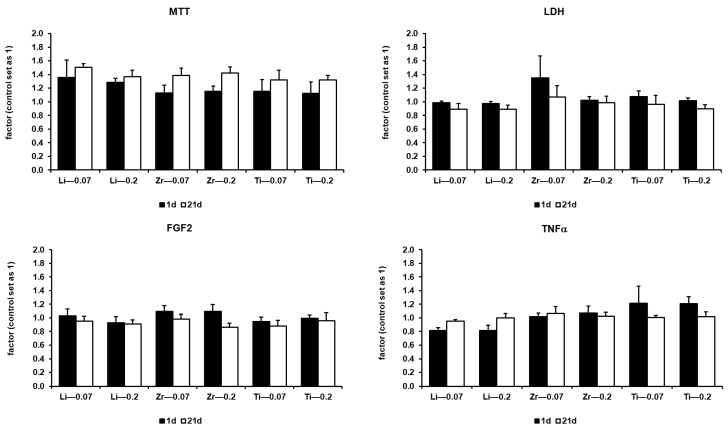
Cell viability of HGF-1 cells growing on different specimen disks at day 1 and day 21. Factors correlated to the control (set as 1) were shown. Proliferation rate was described by an MTT assay (*p* = 0.009); cytotoxicity was described by the release of LDH to culturing media (*p* = 0.009); grade of differentiation potential was described by the expression of FGF2 (significant for Li—0.2, *p* = 0.009, and Ti—0.07, *p* = 0.44); possible inflammation reaction was described by the expression of TNFα (*p* = 0.009–0.014).

**Figure 2 ijms-22-07700-f002:**
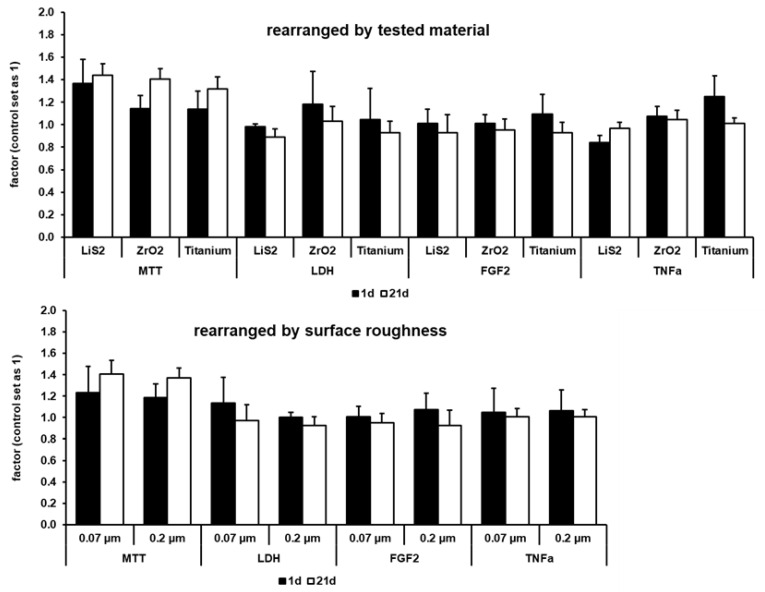
Cell viability of HGF-1 cells growing on different specimen disks at day 1 and day 21, rearranged by tested material (*p* = 0.0002) and surface roughness (*p* = 0.000014). Factors correlated to control (set as 1) are shown. Proliferation rate was described by an MTT assay (*p* = 0.017 for material and *p* = 0.009 for roughness); cytotoxicity was described by the release of LDH in culturing media (*p* = 0.009 for material and for roughness); grade of differentiation potential was described by the expression of FGF2 (significant for titanium group, with *p* = 0.036, and for 0.2 µm roughness group, with *p* = 0.009); possible inflammation reaction was described by the expression of TNFα (significant, with *p* = 0.009, in roughness and significant, with *p* = 0.015, in LiS_2_ and titanium groups).

**Figure 3 ijms-22-07700-f003:**
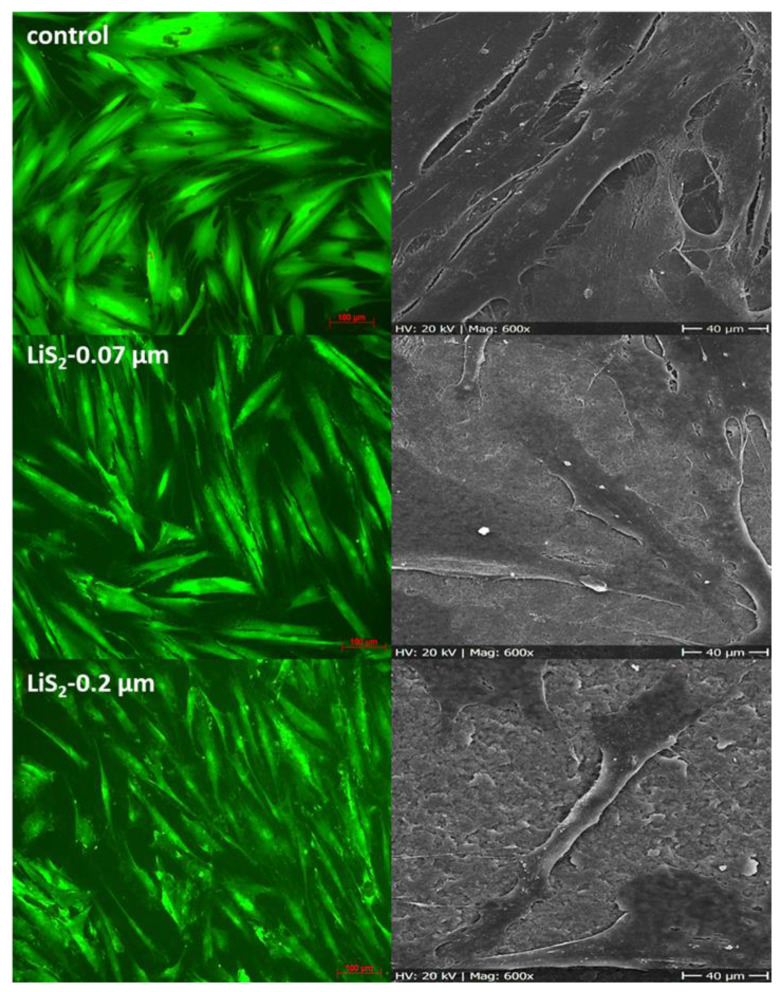
HGF-1 cells growing on lithium disilicate and on a control membrane (ThinCert™) at day 1 (left site = living/dead staining, magnification factor ×100, size bare 100 µm; right site = SEM, magnification factor ×600, size bare 40 µm).

**Figure 4 ijms-22-07700-f004:**
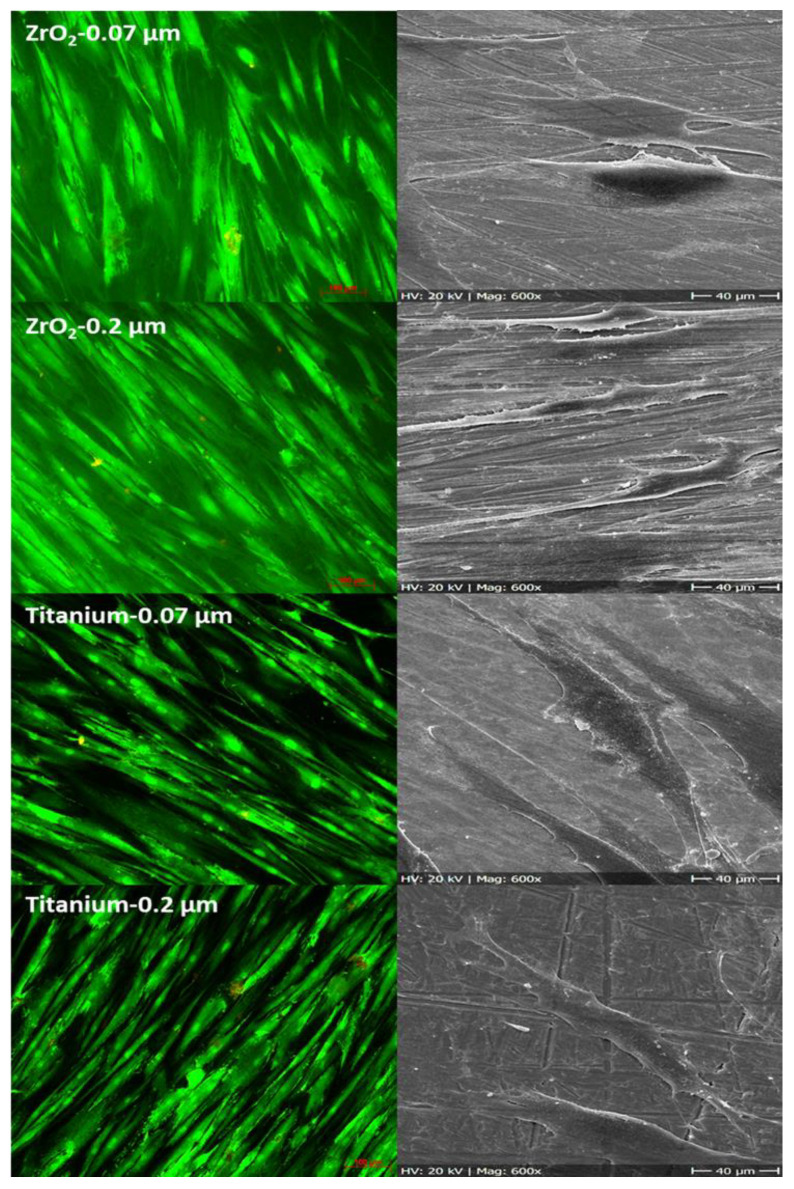
HGF-1 cells growing on zirconia and on titanium at day 1 (left site = living/dead staining, magnification factor ×100, size bare 100 µm; right site = SEM, magnification factor ×600, size bare 40 µm).

**Table 1 ijms-22-07700-t001:** Grinding protocol.

Material	Grinding-Wheel (RA in µM)	Rotation	Suspension	Force (N)	Time (s)
Ti—0.07	MD-Gekko 320 MD-Gekko 1000 MD-Largo	Backwards Backwards Forwards	Water Water 15 µm suspension	40 40 40	120 60 120
Ti—0.2	MD-Gekko 320 MD-Gekko 1000 MD-Gekko 600	Backwards Backwards Forwards	Water Water Water	40 40 50	120 30 15
Zr—0.07	MD-Piano 120 MD-Piano 600 MD-Piano 500	Backwards Forwards Backwards	Water Water Water	90 60 90	240 60 30
Zr—0.2	MD-Piano 120 MD-Piano 220 MD-Piano 220	Backwards Forwards Backwards	Water Water Water	90 90 90	240 120 40
Li—0.07	MD-Piano 120 MD-Piano 500 MD-Piano 600	Backwards Backwards Backwards	Water Water Water	90 90 90	240 90 30
Li—0.2	MD-Piano 120 MD-Piano 220 MD-Piano 220	Backwards Forwards Backwards	Water Water Water	90 90 90	240 90 30

## Data Availability

The datasets analyzed during the current study are available from the corresponding author on reasonable request.

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
