# Peer review of "Biocompatibility of Lithium Disilicate and Zirconium Oxide Ceramics with Different Surface Topographies for Dental Implant Abutments"

_ijms, 2021, doi:10.3390/ijms22147700_

Round 1

Reviewer 1 Report

The study described was conducted at a high methodological level. The authors investigated the use of modern ceramics as materials for the manufacture of abutments (in particular, the possibility of using lithium disilicate for subgingival use).

The authors compared the biocompatibility of that material, which successful application for the supragingival restorations (inlays, crowns) was well known as a classic material, in a new field of application - for the manufacturing of abutments (titanium) and zirconium dioxide.

The only concern is the limited amount of cytokines measured since it is rather difficult to judge the possible presence/absence of an inflammatory reaction based on the determination of only one cytokine. I suggest adding the reference to the studies with similar methodology (where the conclusions were made based on the determination of single cytokines) or discuss that point as the limitation of the study.

Author Response

Responses to Reviewer's Comments:

The only concern is the limited amount of cytokines measured since it is rather difficult to judge the possible presence/absence of an inflammatory reaction based on the determination of only one cytokine. I suggest adding the reference to the studies with similar methodology (where the conclusions were made based on the determination of single cytokines) or discuss that point as the limitation of the study.

We agree with the reviewer. The study is strongly limited by experimental design. To fortify this, we point out that in results and in material and methods.

  1. Results

…For interpretation, strong limitation of the study by chosen TNFa as a single representative for cytokines should be considered…

4.8 Statistical Analysis

…By chosen experimental design and chosen testing parameter, the study was strongly limited…

Reviewer 2 Report

IJMS-1280660 - Biocompatibility of Lithium Disilicate and Zirconium Oxide Ceramics with Different Surface Topographies for Dental Implant Abutments

Abstract

The term "Vitality" is not the best to be used for MTT assay. The authors can change to "Cell metabolism" or "Mithocondrial Metabolism" for better understanding. 

There is no conclusion on Abstract. The sentence "The results may help to gain an understanding of the biological effects of different implant materials and their interaction with the oral tissues" is not a conclusion. It is only a statement of the authors based on the overall findings and not in the study's specific aims. 
Introduction
"After tissue maturation the connective tissue portion, located between the epithelium and the marginal bone, is poor in cells and in vascular structures but rich in collagen fibers. These fibers run more or less parallel to the surface of the abutment." Would you please add references for each statement present in each sentence?
"The second generation of lithium disilicate ceramics (also known as "IPS Empress 2" or "IPS e max" (Ivoclar Vivadent, Germany) )". The second generation of lithium disilicate ceramics is called E-max. Empress 2 is the first generation. Please correct. 
The author could highlight in the Introduction the relevance of using lithium disilicate ceramic as abutments of implants.
Would you please describe the main difference between the present study and the results with similar evaluation already published? The statement "there is scarce evidence on the biological response of the mucosa towards this material" is not a valid justification. 

M&M
Were used primary cells? There is no mention to the Ethics Committee. 
Why use One-way ANOVA? The suitable statistical method for this experiment is a three-way ANOVA, considering the main factors "time of evaluation", "roughness and "material". The authors should change the statistical analysis.
Results
In the results section cannot be noted the comparison among the 3 factors in one figure. Please, change the figures to one with the representation of the results considering different roughnesses and different materials according to the day of evaluation.
Discussion
4th paragraph – The authors should better explain the differences between the studies to justify the different results obtained.
"If lithium disilicate proofs to be an alternative to zirconia, not only regarding its biomechanical properties, but also from a biological standpoint, clinicians could benefit from some advantages of this material. In contrast to zirconia, that can only be processed using a CAD/CAM workflow, lithium disilicate in addition can also be pressed in the desired shape. Moreover, the material has different optical properties [42]." This paragraph has superficial information. Would you please add relevant information? Which advantages could the clinicians be benefited? Both zirconia and lithium disilicate (LD) can be prepared from CAD-CAM. Would you please specify the benefits of LD ceramic on this topic? Have the material different optical properties? Which ones? How can this be relevant?
Would you please standardize first or third person on the writing?
The study did not present any conclusion based on the main purpose. Please write a conclusion at the final of the Discussion or as a separate section. 

Author Response

Responses to Reviewer's Comments:

Abstract:

The term "Vitality" is not the best to be used for MTT assay. The authors can change to "Cell metabolism" or "Mitochondrial Metabolism" for better understanding.

Thank you for your suggestion. We are agree with the reviewer, that the term vitality is misleading. We changed the sentence in the abstract. In the manuscript, we used the term viability. Here we summarized all assay, which has a direct connection to cell vitality.

…Proliferation (MTT), Living/Dead staining, cytotoxicity (LDH), proliferation (FGF2), and inflammation (TNF) were analyzed…

There is no conclusion on Abstract. The sentence "The results may help to gain an understanding of the biological effects of different implant materials and their interaction with the oral tissues" is not a conclusion. It is only a statement of the authors based on the overall findings and not in the study's specific aims.

            The reviewer is right. We changed that paragraph to:

…The results indicate that the two tested high-strength ceramics and surface properties are biologically suitable for transmucosal implant components. The findings may help clinicians to choose the most appropriate biomaterial as well as the most appropriate surface treatment to use in accordance with specific clinical dental applications.

Introduction

"The second generation of lithium disilicate ceramics (also known as "IPS Empress 2" or "IPS e max" (Ivoclar Vivadent, Germany))". The second generation of lithium disilicate ceramics is called E-max. Empress 2 is the first generation. Please correct.

            Thank you for pointing that out. We correted that to:          

…The second generation of lithium disilicate ceramics (also known as “IPS e max” [Ivoclar Vivadent, German] )

Would you please describe the main difference between the present study and the results with similar evaluation already published? The statement "there is scarce evidence on the biological response of the mucosa towards this material" is not a valid justification.

            The Reviewer is right. We have to be more precise in that paragraph. We added:

            …Thus, while the material has been used clinically as an abutment material in close contact with the mucosa for some time now, the number of studies that have investigated its biocompatibility with these specific human cell lines is rather small (36-40).

Furthermore, the influence of surface modifications, which are common for abutments, on the cell response is interesting. That is why it is important to include this factor in the biocompatibility testing. Some studies report of surface modification (37-39), some do not (36,40). 

Only very few studies have compared the new material LS with the common abutment materials titanium and/or zirconium in cell culture, also considering the aspect of surface roughness by testing different surface roughness (38). 

Material and Methods

Were used primary cells? There is no mention to the Ethics Committee.

In our study, we use commercially primary cell line (HGF-1; ATCC® CRL-2014™) supplied by LGC Standards GmbH (https://www.lgcstandards.com/DE/de/HGF-1-Gingival-Fibroblast-Human-Homo-sapiens-/p/ATCC-CRL-2014). There is no need for an ethic approval. To avoid misconceptions, we delete the term “primary”.  

Further, we would never use material or cells isolated from human tissue without permission and ethical approval. 

4.3. Primary Human Cell Culture

Primary Human gingival fibroblasts (HGF-1; ATCC® CRL-2014™; LGC Standards GmbH, Wesel, Germany) were delivered cryopreserved. Thawing of cells was done ac-cording to the manufacturer protocols. Culture techniques of human gingiva fibroblast was performed as described previously [46]. The handling of all human samples strictly adhered to the “Declaration of Helsinki”. 

Why use One-way ANOVA? The suitable statistical method for this experiment is a three-way ANOVA, considering the main factors "time of evaluation", "roughness and "material". The authors should change the statistical analysis.

Chosen study design equates a proof of concept study. As part of good laboratory practices, the cell culture part was repeated three times. In addition, all described analyses by itself were done in triplicates. For this kind of limited study, a statistical analysis of data sets with mean values and standard deviations verified by an additive one-way ANOVA has to be sufficient. With a simple one-way ANOVA, we could not detect stable significance, as we wrote in our results. Our null hypotheses define, that observed changes are false true, if there is no significance with a p-value of at least 0.05.

To substantial results, data were imported in SPSS (version 27). Due to the non-normal distribution of the data and independent parameter, the Kruskal-Wallis test was used. The level of significance was set at p < 0.05. Now, with the Kruskal-Wallis test, we could detect significance between analysis days with a p-value of at least 0.014. Observed changes between day 1 and day 21 were significant with a p-value of 0.009 for MTT and LDH release and with a p-value of at least 0.014 for TNFa expression. There was no significant for FGF2 expression, with exception for LiS2 with a roughness of 0.2 µm and titanium with a roughness of 0.07 µm with a p-value of 0.009 and 0.044 respectively.

Also, significance concerning roughness and material were changed. Concerning roughness, observed changes between day 1 and day 21 were significant with a p-value of 0.009 for MTT, LDH release, and TNFa expression. Observed differences for FGF2 expression, were only significant in 0.2 µm group with 0.009. Concerning material, observed changes between day 1 and day 21 were significant with a p-value of 0.017 for MTT and with a p-value of 0.009 for LDH release. For FGF2 expression, observed changes only in titanium group were significant with a p-value of 0.036. For TNFa expression, observed changes were significant in LiS2 group and titanium group with a p-value of 0.009 and not in ZrO2 group. 

Abstract:

… Statistical analysis was performed by Kruskal-Wallis test. The level of significance was set at p < 0.05. one-way ANOVA and a modified Levene test with a statistical significance at p= 0.05.

 2.1. Cell Viability

Cell viability among tested specimens at day 1 and at day 21 is shown in Figure 1. Changes between day 1 and day 21 were significant with a p-value of 0.009 for MTT and LDH release and with a p-value of at least 0.014 for TNFexpression. There was no significant for FGF2 expression, with exception to LiS2 with a roughness of 0.2 µM and titanium with a roughness of 0.07 µm with a p-value of 0.009 and 0.044, respectively. One-Way ANOVA was performed and no siginficant p-values occurs for assays shown in Figure 1.

Figure 1. Cell viability of HGF-1 cells growing on different specimen disks at day 1 and day 21. Factors correlated to control (set as 1) were shown. Proliferation rate was described by an MTT assay (p = 0.009), cytotoxicity was described by release of LDH to culturing media (p = 0.009), grade of differentiation potential was described by expression of FGF2 (significant for Li-0.2 p = 0.009 and Ti-0.07 p = 0.44), and possible inflammation reaction was described by expression of TNFa (p = 0.009 - 0.014).

2.2 Data analysis with focus on roughness and material

Figure 2, results were rearranged and sorted according to the tested material and surface roughness.

Here, differences in cell viability just influenced by tested implant material or by surface roughness should be carved out.

Considering surface roughness and used material, results were rearranged and summarized results in figure 2. Concerning material, changes between day 1 and day 21 were significant with a p-value of 0.017 for MTT and with a p-value of at least 0.009 for LDH release. For FGF2 expression, observed changes only in titanium group were significant with a p-value of 0.036. For TNFa expression, observed changes were significant in LiS2 group and titanium group with a p-value of 0.009 and not in ZrO2 group. Concerning roughness, changes between day 1 and day 21 were significant with a p-value of 0.009 for MTT, for LDH release and for TNF expression. Observed differences for FGF2 expression, were only significant in 0.2 µm group with 0.009. One-Way ANOVA was performed. Results were significant with a p-value of 0.0002 and 0.000014, respectively.

Figure 2. Cell viability of HGF-1 cells growing on different specimen disks at day 1 and day 21 rearranged by tested material (p = 0.0002) and surface roughness (p = 0.000014). Factors correlated to control (set as 1) were shown. Proliferation rate was described by an MTT assay (p = 0.017 for material and p = 0.009 for roughness), cytotoxicity was described by release of LDH to culturing media (p = 0.009 for material and for roughness), grade of differentiation potential was described by expression of FGF2 (significant for titanium group with p = 0.036 and for 0.2 µm roughness group with p = 0.009), and possible inflammation reaction was described by expression of TNFa (significant with p = 0.009 in roughness and significant with p = 0.015 in LiS2 and titanium group).

4.8. Statistical Analysis

The statistical software SPSS version 27 (IBM, Ehningen, Germany) was used for statistical analysis. By chosen experimental design and chosen testing parameter, the study was strongly limited. Due to the non-normal distribution of the data, independent parameter and small sample size, statistical analysis was one by using the Kruskal-Wallis test was used. The level of significance was set at p < 0.05.

Results

In the results section cannot be noted the comparison among the 3 factors in one figure. Please, change the figures to one with the representation of the results considering different roughnesses and different materials according to the day of evaluation.

Thank you for your suggestion. Nevertheless, I am confused about your request. In Figure 1, we presented data for each comparing material and roughness separately. In Figure 2, we rearranged given data considering roughness and material. We presented in figure 1 as in figure 2 to control normalized factors. In figure 1, we showed four different diagrams. In figure 2, we summarize diagrams, in one according to roughness and in one according to material. Therefore, we could save space for one and presented data considering roughness or material could compare easier. It is an admissible way to consolidate data.

To reach a better clarity, we added another subitem in results.

….2.2 Data analysis with focus on roughness and material

Figure 2, results were rearranged and sorted according to the tested material and surface roughness. Here, differences in cell viability just influenced by tested implant material or by surface roughness should be carved out.

Considering surface roughness and used material, results were rearranged and summarized results in figure 2.

Discussion

4th paragraph – The authors should better explain the differences between the studies to justify the different results obtained.

We added more information in that paragraph and explained the issues more in detail:

…As described in the introduction only very few studies have compared Lithiumdisilicate with the common abutment materials titanium and/or zirconium in cell culture, also considering the aspect of surface roughness (38).

One of the few that included this new material is a cell culture that used single-cell force spectroscopy to evaluate cell adhesion of living gingival fibroblasts on different materials and surface roughness [18]. The authors did not find any difference between the materials, but found significant differences between the different surface roughnesses showing the strongest cell attachment on machined like surface roughness. Due to the different methodology these findings can not be compared with the results of the present study.

Another cell culture experiment compared the biological response of epithelial tissue cultivated on lithium disilicate and zirconia, yielding to assess the influence on gingival would closure of these materials (38). The authors looked at cell migration and found the best migration on Zirconia. Cell adhesion was better on rough and polished Lithiumdisilicate than on the zirconia specimen. Increasing surface roughness lead to better cell adhesion. However, the roughness of the specimen varied from Ra 0.01 µm to 2.53 µm. Moreover, the experiment used a chicken epithelium model and not human gingivofibroblasts. Hence, these findings can not be compared. 

"If lithium disilicate proofs to be an alternative to zirconia, not only regarding its biomechanical properties, but also from a biological standpoint, clinicians could benefit from some advantages of this material. In contrast to zirconia, that can only be processed using a CAD/CAM workflow, lithium disilicate in addition can also be pressed in the desired shape. Moreover, the material has different optical properties [42]." This paragraph has superficial information. Would you please add relevant information? Which advantages could the clinicians be benefited? Both zirconia and lithium disilicate (LD) can be prepared from CAD-CAM. Would you please specify the benefits of LD ceramic on this topic? Have the material different optical properties? Which ones? How can this be relevant?

Thank you for your suggestion, however we find it confusing. In the paragraph you pointed out, we wrote that -in contrast to Zirconia - LD can be pressed, hence fabricated with a totally different work flow than Zirconia, leading to a different approach that was already described in the introduction. We think that this is a distinct difference. Moreover, we in deed pointed out that the material has different optical properties.

To make the facts even clearer we added the following sentence:

…As described in the introduction, these material characteristics lead to a novel approach for the fabrication of implant borne supra-structures, yielding in a screw-retained restoration without a traditional abutment, which simplifies the treatment.

Would you please standardize first or third person on the writing?

Thank you for your kind reminder. We reviewed the manuscript carefully, but it was hard to find something. Maybe we have blinders on. A sample given by the reviewer would be helpful.  However, we assume that the reviewer means the following sentence. We changed it.

… The two different surface roughness that we were tested in this experiment did not show to have any effect on HGF-1. … However, in this particular experiment we chose two surfaces below the threshold of 0.2 µm were chosen [6,7].

The study did not present any conclusion based on the main purpose. Please write a conclusion at the final of the Discussion or as a separate section.

            We added a conclusion as a final remark to the Discussion:

…..From this in vitro experiment, it may be concluded that regarding biocompatibility and cell response, the two tested high-strength ceramics and surface properties are biologically suitable for transmucosal implant components. However, randomized clinical trials have to proof the soft tissue response and clinical performance of lithium disilicate material in the transmucosal area of implant restorations over time.

Round 2

Reviewer 2 Report

Overall, the review was well done. The advantages to using LS than zirconia as implant abutment can be addressed to the better bonding to the first ceramic, in cases of mixed esthetic procedures, for example. However, there is no need to add this information to the text.

Thank you for the review. The points were adequately revised.